# Hospitalized Adults’ Willingness to Use Mobile Apps for Air Quality and Heat Monitoring: A Survey-Based Study

**DOI:** 10.3390/ijerph22111733

**Published:** 2025-11-16

**Authors:** Elizabeth Cerceo, Lydia Abbott, Roger Sheffmaker, Mariam Ansar, Jean-Sebastien Rachoin, Katherine T. Liu

**Affiliations:** 1Department of Medicine, Cooper University Healthcare, Cooper Medical School of Rowan University, Camden, NJ 08103, USA; mariamansar98@gmail.com (M.A.); rachoin-jean@cooperhealth.edu (J.-S.R.); 2Cooper Medical School, Rowan University, Camden, NJ 08103, USA; sheffm17@rowan.edu; 3School of Medicine, Tufts University, Boston, MA 02111, USA; lydia.abbott@mainehealth.org; 4Department of Medicine, MaineHealth Maine Medical Center, Portland, ME 04102, USA; katherine.liu@mainehealth.org

**Keywords:** climate health, environmental health, air quality, air pollution, extreme heat, applications

## Abstract

Climate change and environmental degradation pose growing threats to health. Despite increasing recognition of these risks, climate-related education and counseling are rarely incorporated into adult inpatient care. A survey-based study was conducted with 250 adult inpatients on the medicine services at Cooper University Health Care (New Jersey) and Maine Medical Center (Maine). Patients received a standardized 30-s educational statement from their physician on the health impacts of air pollution and extreme heat, with introduction to two smartphone applications on air quality and heat conditions. Survey items evaluated patients’ prior awareness of environmental health risks, willingness to use digital monitoring tools, and perceived barriers to use. Descriptive statistics and content analysis were used for data interpretation. Overall, 84% of participants reported awareness of environmental threats to health, indicating high baseline recognition. However, only 50% expressed willingness to adopt smartphone apps as protective tools with barriers including lack of smartphone access, unfamiliarity with technology, and concerns about utility during hospitalization. Twenty-three percent of participants in Maine did not own a smartphone, as compared with 7% in NJ. Despite less smartphone ownership in Maine compared to NJ, participants showed similar willingness to use the suggested apps for monitoring environmental conditions (53% vs. 49.3%). Responses suggested that while patients generally acknowledge climate-related health risks, enthusiasm for technological solutions varies considerably, especially among older and underserved populations. This study highlights a critical gap between awareness of climate health risks and the adoption of digital health tools for self-protection. While brief inpatient education may increase recognition, technology-based interventions alone may not reach all patient groups. Future strategies should focus on accessible, low-barrier methods of environmental health education in clinical care, including integration into inpatient counseling and discharge planning. Addressing technology access gaps and tailoring resources to diverse populations will be essential for advancing climate-related patient education in healthcare settings.

## 1. Introduction

Climate change and environmental degradation are increasingly recognized for their profound impact on human health. Climate-related factors, such as air pollution and extreme heat, are particularly well-documented in relation to cardiopulmonary conditions, heat-related illnesses, and other health concerns and are associated with increased morbidity and mortality [1,2,3,4]. Heat-related deaths are estimated to be around 12,000 annually, comparable with gun homicides [5]. Air pollution, specifically fine particulate matter (PM_2.5_), has emerged as the greatest global risk for disease burden [1]. Estimates of deaths from fossil fuel-induced air pollution range from 8.9–14.5 million annually [6,7,8,9]. Inflammation from systemic exposure to these pollutants trigger cardiovascular disease [2,10], pulmonary disease [11,12,13], cancer [14,15], and stroke [16,17,18]. Now, associations with dementia [17], Parkinson’s disease [19,20], chronic kidney disease [21,22,23], and diabetes [9,24,25]. In the hospitalized populations, considerations of the patients’ conditions upon discharge may predict readmission [26,27,28,29,30,31]. Despite mounting evidence linking climate change to adverse health outcomes, physicians rarely counsel patients on the health risks associated with climate change, a gap highlighted in previous studies [32].

Physicians hold a trusted position and have the potential to educate patients on preventive measures regarding environmental risks [33,34,35]. However, many clinicians avoid the topic, often perceiving it as complex with few immediate solutions and politically sensitive for some patient populations. While pediatricians have demonstrated success in integrating climate health education into well-child visits, as evidenced by positive results in parental perceptions, there remains a paucity of data concerning adult inpatient perceptions of climate-related health risks [36,37,38,39]. Certainly, patients should be counseled on environmental health impacts in the outpatient setting as well but a hospitalization may serve as a “teachable moment” when patients may be receptive to climate and environmental health education while in the hospital [39,40]. Additionally, mobile applications (apps) may be an effective means to monitor and mitigate environmental harms. Apps designed to convey environmental information can also influence health-related behaviors [41].

This study assessed hospitalized adults’ perceptions of climate change as a health risk and their awareness of environmental factors affecting their well-being. Additionally, this study explores patients’ interest in using technology, such as smartphone applications, to monitor environmental risks and support personal health management. Understanding patient perspectives on climate-related health threats can inform strategies to integrate climate health discussions into routine inpatient care, enhance patient education, and foster greater engagement with preventive environmental health measures.

## 2. Methods

This prospective, survey-based study targeted adult inpatients at two major healthcare institutions: Cooper University Health Care in Camden (New Jersey) and Maine Medical Center (Maine). Participants were identified using a convenience sampling method from among patients admitted to the adult inpatient medicine service. Recruitment was coordinated to align with the availability of attending physicians and medical students, who approached eligible patients during their clinical shifts to introduce the study and obtain consent. This study was deemed exempt from review by the MaineHealth Institutional Review Board (study #2168235-2) and was classified as quality improvement at Cooper University Health Care, exempting it from formal IRB approval.

### 2.1. Participants

A total of 250 inpatients were enrolled between November 2023 and September 2024, with 149 participants from Maine Medical Center and 101 from Cooper University Health Care. Participants were ≥18 years of age and admitted with a primary or secondary diagnosis of cardiopulmonary or heat-related illness. Patients who were critically ill, unable to provide responses (e.g., due to delirium or dementia), or had unrelated medical conditions were excluded. The sample size was based on prior patient assessments of climate health perceptions, in which 234 families were interviewed [36]. Sampling was clustered based on the research team’s availability. The two sites were selected to increase geographic variability and more diverse patient representation.

### 2.2. Study Encounter

Potential participants were approached separately after all clinical care was delivered. They were provided with a patient information sheet explaining the study and its potential risks and benefits. Verbal consent was obtained. The authors delivered a 30-s educational statement to patients about the impact of environmental factors like air pollution and extreme heat on their health, followed by a verbal introduction to two smartphone applications (EPA AIRNow version 4.3.1 and OSHA-NIOSH Heat Safety Tool version 3.7) for monitoring environmental conditions. These applications were selected because they are government-supported, free, and easy to use. Both offer color-coded information on health risks from heat and poor air quality. The standardized educational statement was developed based on the pediatrics climate change counseling study by Lewandowski et al. In that study, environmental health risks were placed alongside other pediatric preventative health behaviors like wearing seatbelts and helmets when riding bikes [36]. Thus, for a study in adults, counseling on smoking avoidance aligned with an example of preventative measures that would be relatable to most patients. The survey questions were developed by the authors and beta-tested with physicians outside the study with climate health expertise. The questions were kept intentionally short, in recognition of the acuity of patients’ condition in the hospital setting. If patients were interested in using the applications, the clinicians conducting the study helped the patients/families download them on their smart phones in real time. Responses were collected immediately after the intervention. Qualitative comments were recorded by the healthcare professional interviewing the patient.

### 2.3. Researcher-Delivered Statement and Survey

“Many health problems like heart and lung disease can be worsened by environmental conditions such as air pollution and heat waves (*if in summer*). Just as I counsel patients not to smoke, I also counsel my patients on the importance of protecting themselves from air pollution and during dangerous heat waves. There are apps for air quality (EPA AIRNow) [*and heat* (*OSH-NIOSH Heat Safety Tool*)]. It is important to try to stay indoors when the air quality is very bad (*and to be out of the heat when it is dangerously hot*).”

Survey:

Were you previously aware that air pollution and extreme heat could harm your health? Yes/No

Do you think you would use this (these) app(s) to protect your health? Yes/No/No smart phone

May I help you download a free, government-produced air quality monitoring app on your phone? Yes/No/No smart phone

Do you have any thoughts you would like to share?

### 2.4. Data Analysis

#### 2.4.1. Descriptive Statistics

Categorical data were presented as frequency and percentages. Chi-square tests were utilized to evaluate the relationship among categorical variables, with a *p*-value less than 0.05 considered statistically significant.

#### 2.4.2. Interpretive Content Analysis

Interpretive content analysis, a hybrid approach that combines elements of traditional content analysis with interpretive techniques to understand underlying meanings, contexts, and assumptions, was chosen to categorize content while interpreting latent meanings and cultural or contextual significance. This more interpretive approach was well-suited to the brief, open-ended patient responses, allowing for the extraction of meaningful insights despite limited narrative depth.

The interpretive content analysis used in this study was primarily inductive, aiming to generate themes from the qualitative data rather than test a predefined framework. The approach was not informed by a specific theoretical lens, though it was guided by the overarching research question regarding patient perceptions of climate-related health risks and the acceptability of mobile health interventions. While the survey included both quantitative and qualitative components, the qualitative analysis was conducted independently of the quantitative findings to ensure that emergent themes were derived from participant narratives rather than structured item responses.

Data collection and qualitative analysis occurred in a sequential manner. All qualitative responses were collected first, followed by the thematic coding process. Coding was conducted by multiple members of the research team who independently reviewed responses to identify recurring concepts. These concepts were then compared, refined through team discussion, and grouped into themes using an iterative process. While a formal coding software was not used due to the brevity of the comments, all responses were analyzed and categorized using a consistent coding matrix to ensure transparency and reproducibility. This analytic strategy was chosen to capture nuanced insights in the patients’ own language while acknowledging the interpretive nature of the data.

Of the 251 participants enrolled, one record was discarded due to a duplicate entry. A total of 113 participants (45%) provided additional comments, 69 at the Maine site and 44 at the New Jersey site. These comments were analyzed for general content rather than verbatim responses. Two researchers (LA and KL) independently coded each comment to identify key themes. Following initial coding, the codes were compared, consolidated, and refined, using interpretive content analysis [42]. Similar codes were merged, resulting in a codebook with 16 unique themes, each with a defined description.

A second round of coding was then conducted (LA, KL, EC), applying the codebook to comments from both sites, followed by one final round of discussion and refinement of the codebook to ensure consistency. Multiple codes were applied to each comment where applicable. Negotiated agreement, as previously described by Garrison et al. 2006, was undertaken to reach consensus on final codes [43]. Consensus was reached for each comment. The final codes were used to calculate frequency counts for each theme, categorized into three subgroups (Yes/No/No Smartphone) based on participant responses to a specific survey question, referred to as “may I help you download” in future references.

#### 2.4.3. Researcher Reflexivity

In this study, qualitative data were collected through brief, open-ended patient responses recorded by healthcare professionals during inpatient encounters. As such, researcher reflexivity was critical throughout both data collection and analysis. The interdisciplinary research team included hospitalists, internal medicine residents, and medical students, each of whom brought distinct perspectives shaped by clinical experience, education level, and personal engagement with environmental health. The authors acknowledge that their professional roles as physicians and medical trainees may have influenced both the delivery of questions and the interpretation of patient responses. Efforts were made to minimize this bias by employing a standardized script for all patient encounters and through immediate recording of patient comments to reduce recall distortion. All researchers involved in qualitative coding engaged in reflexive discussions to acknowledge their assumptions, values, and potential influence on thematic interpretation. Coding was conducted independently by multiple reviewers and finalized through consensus, using negotiated agreement to reduce individual coder bias.

#### 2.4.4. Potential Bias

This study used a convenience sample, thus inherent bias is present. As participants were hospitalized, they may overrepresent those with higher disease burdens or poorer management, especially among marginalized populations. Not all eligible patients were enrolled due to researcher availability within the study dates. Courtesy bias may also have influenced responses, as patients could have answered in ways they perceived the researcher preferred.

## 3. Results

### 3.1. Descriptive Statistics

A total of 250 adult inpatients participated in the survey, with 149 participants from Maine Medical Center and 101 from Cooper University Health Care. Out of the 250 respondents, 212 (84%) reported being aware of the link between environmental factors such as air pollution and extreme heat and their health prior to the educational intervention. However, there was significant variation in patients’ willingness to use smartphone applications to monitor environmental conditions. Of the 250 respondents, 124 (50%) indicated they would likely use the apps provided to protect their health. Among those who chose not to use the apps, several reasons were cited, including lack of access to a smartphone, unfamiliarity with technology, and a general belief that the apps were unnecessary for their health management (Table 1).

### 3.2. Site Differences

Participants at both study sites demonstrated awareness of the link between environmental factors, such as air pollution and extreme heat, and their health, with a total of 212 participants (84%) reporting being aware of this connection. However, there were notable differences between the sites. In Maine, 90% of participants reported awareness of this link, while only 77% of participants at the New Jersey site reported similar awareness. Twenty-three percent of participants in Maine did not own a smartphone, as compared with 7% in NJ. Despite less smartphone ownership in Maine compared to NJ, participants showed similar willingness to use the suggested apps for monitoring environmental conditions (53% vs. 49.3%) (Table 2).

A chi-square test comparing prior knowledge (“Were you previously aware…”) with patients’ perception of their likelihood to use the app (“Do you think you would use…”) and their willingness to accept help installing the technology-based tool on their phone or tablet (“May I help you download…”) revealed a significant overall association (*p* = 0.001 and *p* = 0.004, respectively). However, when analyzing responses by location, no significant differences were observed at the Maine site (*p* = 0.457 and *p* = 0.685), suggesting that prior awareness did not influence responses in this group. In contrast, the New Jersey site showed a strong association between prior knowledge and both likelihood of use (*p* < 0.001) and willingness to accept help (*p* < 0.001), indicating that awareness played a more significant role in this population. These findings suggest that prior knowledge may not uniformly impact patient engagement but could be a relevant factor in specific settings.

### 3.3. Interpretive Content Analysis by Acceptance of Technology-Based Tool 

Codes were organized by response to the question “may I help” and reviewed for frequency of code occurrence. (Table 3, Appendix A) Comments from those who did not have smartphones were coded alongside the other comments. Major themes and the relationships of these themes to the acceptance of a technology-based tool for participants with smartphones are listed below. In total, 113 total comments were collected from 250 participants, with 49 comments for the ‘yes’ group, 47 for the ‘no’ group, and 17 for the ‘no smartphone’ group.

#### 3.3.1. Application as a Potential Resource or Tool

Among those who accepted the tool, the applications were overall seen as an important resource. Some participants indicated that they would use the resource to directly guide their behavior, with comments such as “I am not super tech literate, but I think I’ll try it especially with my breathing how it is. Every little bit may help, and it won’t hurt.” Others noted specific health concerns that the applications could address, “[My] breathing is worse when [there is] dust and pollution around.” Among those who declined assistance, they often noted alternative resources for environmental air quality and heat index information, such as the news or radio.

#### 3.3.2. Technology as Both a Tool and Barrier

The idea of technology as a tool and a barrier was raised directly by patients, particularly in the group with smartphones who did not accept the tool, but also generally observed by the researchers. Among those who did not accept the tool, discomfort with technology was often mentioned, with patients by stating that they don’t typically use apps, dislike having things on their phone, or are not very “technologically savvy.” Additionally, technology posed a barrier for those who accepted the technology-based tool due to issues such as forgotten app store passwords preventing the downloading of the app, despite the participants’ acceptance of help. Finally, a notable number of patients, as mentioned earlier, did not own a smartphone themselves.

#### 3.3.3. Concerns Around Environmental and Climate Health Impacts

Within each group, some participants directly acknowledged climate change impacts on their personal health within their comments, such as “I never go out when the air quality is bad. My COPD gets really bad.” Others expressed concerns about additional environmental health issues, such as radon or PFAS chemicals. One participant specifically questioned how we, as a broader community, can focus on one issue, such as air quality, when there are so many interconnected factors at play.

#### 3.3.4. Challenge of Balancing Risk and Desired Behavior

Patients who declined help downloading the app or did not own a smartphone acknowledged the challenge of balancing their desired behaviors with the need to protect themselves from environmental factors, such as air pollution and heat. One patient specifically noted being unable to call out of work due to the air quality, stating they would not check this information because of it. Additionally, some patients with smartphones who did not download the app reported already modifying their behavior, such as staying indoors, limiting outdoor time, or using air conditioning. As noted earlier, many participants with smartphones who declined the tool indicated they were receiving environmental information from other sources. 

#### 3.3.5. Concern for the Health and Wellness of Others

Among those who accepted help, there was a greater overall awareness of the health and wellness of others, including the intention to share with others, recognition of vulnerable subpopulations, and a need for increased awareness for these tools in the broader community. Patients who expressed an intention to share often mentioned a specific person they thought would benefit from or be interested in the tool. Several participants also noted a lack of awareness in the general population and emphasized the need for this information to be more widely accessible.

## 4. Discussion

As the first study of adult inpatient climate perceptions, the results highlight that around half of hospitalized patients acknowledge awareness of environmental health impacts from air pollution and extreme heat. While most respondents acknowledged the climate–health connection, far fewer expressed willingness to engage with smartphone applications for environmental monitoring. Common barriers included lack of smartphone access, discomfort with technology, and the acute focus on immediate health issues during hospitalization. This reluctance to use technology was even more pronounced in Maine, where although 90% of respondents recognized the health effects of climate change, only 40% were willing to accept assistance in downloading the apps. A contributing factor may be the 23% of participants in Maine who lacked a smartphone, compared to only 7% in New Jersey. However, in both groups, few patients were willing to have the apps downloaded by clinicians, suggesting that those who are most likely to benefit from these tools may also be the least inclined to engage with them.

The two study sites exhibited distinct characteristics that may have influenced these findings. Camden, NJ, has a more urban and disadvantaged population compared to Maine, which may help explain some of the differences in awareness and smartphone usage. As a disadvantaged community with a poverty rate of 29.8%, patients in Camden NJ may be particularly sensitive to environmental stressors, especially when ill [44]. Across both sites, barriers included lack of smartphone access, discomfort with technology, and skepticism that such apps would meaningfully change behavior. Ensuring digital health equity is essential, as individuals from lower socioeconomic backgrounds continue to face a persistent “digital divide” and significant barriers to adopting new technologies [45,46]. Complacency in assuming widespread access neglects the reality that older adults, particularly those from economically challenged backgrounds, exhibit significantly lower engagement with digital health tools [47,48].

The qualitative comments were particularly instructive in shedding light on patient attitudes. While patients recognized the health consequences of climate change, many also expressed helplessness about solutions. For some, acute illness limited engagement with preventive discussions, reinforcing the importance of exploring perceptions in outpatient or community settings, where patients may be more receptive [36]. While most comments were positive regarding the recognition of climate change and its health impacts, the study did not assess the depth of patients’ understanding nor their current preventive actions.

These results align with national trends, where a majority of Americans believe global warming is occurring, but suggests that those experiencing acute health issues may be more attuned to additional threats to their health [49]. In Camden NJ, 77% of hospitalized patients acknowledged the health impacts of environmental factors, which, while lower than the 90% in Maine, is still higher than in national studies.

To reach vulnerable patients, complementary strategies beyond technology are essential. Printed handouts, verbal counseling during hospital stays, and traditional channels such as television, radio, or community-based outreach may be more effective. Such materials could serve as accessible alternatives to smartphone applications. In addition, integrating environmental risk checks into routine discharge planning alongside medication and follow-up instructions may help ensure that patients leave the hospital with actionable, personalized guidance to protect their health. Strengthening public health warning systems for heat and poor air quality, and ensuring these alerts are disseminated in patient-centered ways, could further enhance the effectiveness of these inpatient interventions.

Limitations of the study include relatively small sample size drawn from health systems in the Northeast region of the US. While the characteristics of the two sites differed in terms of urban and rural populations, they both lie in liberal-predominant states. Inclusion of a more diverse geographic mix may have produced different results. Additional limitations include the lack of demographic data for participating patients. Keeping responses anonymous without identifying information allowed the elimination of informed consents which would have added significant time to the encounter and may have been more burdensome for hospitalized patients. The questions were kept very brief since respondents were hospitalized for acute conditions. However, a longer questionnaire may provide finer discrimination on patient perceptions. The survey questions developed by the researchers are not formally validated tools and future iterations of surveys may better reflect patient perceptions.

The authors acknowledge the potential for self-selection bias in this study. Individuals who were already more environmentally concerned or interested in climate-health issues may have been more likely to participate in the survey or provide favorable responses, which could have resulted in an overestimation of awareness or willingness to engage with the proposed tools. Participation was voluntary, brief, and unrelated to clinical care, and all eligible inpatients approached during the study period were included regardless of baseline views. The authors also recognize that interviewer–patient dynamics may have introduced bias, particularly as physicians or medical trainees conducted the surveys. Patients may have felt inclined to respond in a socially desirable manner or provide answers they perceived as preferred by the interviewer. To mitigate this, a standardized script was used for all encounters, responses were recorded immediately to reduce interpretation bias, and multiple coders independently reviewed qualitative comments to minimize individual influence. Relatively few patients (113 of 250) provided qualitative comments, so qualitative assessment was drawn from a smaller pool of patient experiences. As a result, the qualitative themes highlight illustrative perspectives and patterns of response, but are not intended to be interpreted in a quantitative or generalizable manner.

Future studies should build on these findings, examining the long-term impact of these interventions, assessing whether follow-up education leads to sustained behavior changes, such as increased use of climate-related resources or other preventive measures, ultimately enhancing health outcomes in the face of climate-related risks. Larger, multisite studies across diverse regions could better capture demographic and geographic variation in perceptions. Prospective studies in outpatient and community settings may provide deeper insight into patient receptiveness when not acutely ill. Mixed-methods approaches combining validated surveys with richer qualitative interviews would help assess both the depth of understanding and the barriers to behavioral change. Intervention trials could evaluate the effectiveness of different risk-communication strategies ranging from clinician-delivered counseling and printed resources to digital tools to improve patient knowledge, preparedness, and resilience. Finally, research should assess whether integrating climate-health education into inpatient and outpatient workflows leads to sustained engagement with preventive measures and measurable clinical health outcomes.

## 5. Conclusions

This pilot study demonstrates that brief educational interventions in the inpatient setting can raise awareness and initiate important discussions about climate health. A significant proportion of adult inpatients recognize the health risks associated with climate change, particularly in relation to air pollution and heat, often with an acute awareness of their personal vulnerability. However, while smartphone apps and other digital tools present potential opportunities for education and engagement, their effectiveness may be limited among certain patient populations, such as older adults or those with restricted access to technology. By clarifying patient needs, testing innovative communication strategies, and embedding climate-health education into routine practice, future work can ensure that healthcare systems not only acknowledge climate risks but actively equip patients, especially the most vulnerable, to protect their health in a changing environment. Equipping clinicians with the necessary tools and knowledge to engage in climate health discussions is vital for improving patient understanding and preventive actions.

## Figures and Tables

**Table 1 ijerph-22-01733-t001:** Survey results.

Question	Yes	No	No Smart-Phone
Were you previously aware that air pollution and extreme heat could harm your health?	212	38	n/a
Do you think you would use these apps to protect your health?	80	127	43
May I help you download a free, government-produced air quality monitoring app on your phone?	104	103	43

**Table 2 ijerph-22-01733-t002:** Site differences.

Question	Camden, NJ	Portland, ME	Significance
Were you previously aware?	Yes 77% No 23%	Yes 90% No 10%	*p* = 0.007
Would you use an app?	Yes 53% No 40% No cell phone 7%	Yes 49.3% No 27.3% No cell phone 23.3%	*p* = 0.002
May I help download it?	Yes 45.5% No 47.5% No cell phone 7%	Yes 39.3% No 37.3% No cell phone 23.3%	*p* = 0.007

**Table 3 ijerph-22-01733-t003:** Qualitative comments.

Theme	Example
**In favor of apps**	
Recognition of personal health harms	“I can’t go outside when pollution is bad. I have air filters.” “My asthma flares up with the weather.” “My asthma is worse when the grass is cut.” “My breathing gets worse when dust and pollution is around.” “I get migraines when the temperature changes rapidly.” “I feel dizzy when there is bad air quality outside.” “I feel like I’m going to faint from the heat.” “I am not super tech-literate but I think I’ll try it, especially with my breathing how it is. Every little bit can help and it won’t hurt.”
Recognition of harms and concurrent risk avoidant	“I hear about it all the time. At my age, there is nothing I can do about it but I know that those who know about it are doing something and I hope they do enough. I stay inside most of the time unless I am going somewhere and I have an air conditioner where I live.” “I use the air conditioner and stay inside when it’s too hot for me.”
General	“This is really helpful!” “This could help me know when to wear a mask.” “I really believe in climate change and the way it affects health. This is very cool and good to know. My neighbor is very into climate change and I will definitely share this with her too.” “More people should be aware of this. I tell my family members to be careful of the heat especially.” “Very cool, I look at this in the weather app but this is more information.” “This is great, now I can know. I knew before but didn’t do anything about it.” “People aren’t taking this seriously enough. You are researchers. What are you doing in terms of legislative advocacy for changes to air quality?”
**Against apps**	
Technological limitations	“I don’t look at my phone every time I go out.” “I don’t like things on my phone, but if you have a piece of paper with this information I would like that.” “I have too many apps already.” “I usually get pollution updates from the news.” “I don’t really use apps so I wouldn’t use this one, but I watch the news and trust that and follow what it says to protect myself.” “I don’t think people look at an app for this information.” “I’m not good with technology.” “We’re too old at this point and not technologically savvy but we know.”
Perceived lack of utility of apps	“I believe God controls the weather.” “I can tell when the weather is bad.” “It’s climate change. That’s all there is to it.” “I caused most of my lung and heart issues through smoking more than air pollution.”
Perceived lack of impact	“I’m not bothered by heat or air.” “I was a farmer and around heat and bad air my whole life without trouble.” “I already do things to maintain my air quality like growing plants for cleaning the air in bedrooms, having salt lamps as well as always having air purifiers running.”

## Data Availability

The original data presented in the study are openly available in Appendix A.

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
