# Peer review of "Hospitalized Adults’ Willingness to Use Mobile Apps for Air Quality and Heat Monitoring: A Survey-Based Study"

_ijerph, 2025, doi:10.3390/ijerph22111733_

Round 1

Reviewer 1 Report

Comments and Suggestions for Authors

This is a survey-based study with 250 adult inpatients on the medicine service, and physicians delivered a standardized 30-second educational statement on the effects of air pollution and heat on health, followed by a brief survey to be completed by patients. 

First of all, the abstract is poorly written and should be revised.

The citation formats are also incorrect. Please double-check with the journal guidelines.

Also, the survey questions are not presented in this study. Can the authors explain why this is not provided?

A sample survey should be presented in the main text or in supplementary materials as a reference.

Also, the limitation of the study is not provided in the discussion or conclusion section. 

The authors claimed "A significant proportion of adult inpatients recognize the health risks associated with climate change, particularly in relation to air pollution and heat, often with an acute awareness of their personal vulnerability. However, while smartphone apps and other digital tools present potential opportunities for education and engagement, their effectiveness may be limited among certain patient populations, such as older adults or those with restricted access to technology".

This claim seemed to be ordinary and common sense. Can the authors explained the novelty of their study?

This work seemed to be brief and not very comprehensive. With only 250 samples, can the authors explain if it is enough to conclude this study?

Comments on the Quality of English Language

The authors should avoid using first-person tone (such as we) in scientific writing. Also, PM2.5 should be written using a subscript.

Author Response

Reviewer 1:

Comment 1: First of all, the abstract is poorly written and should be revised.

Response 1: We have rewritten the abstract so it is now in a more structured format, rather than a single paragraph with the subsections combined.

Comment 2: The citation formats are also incorrect. Please double-check with the journal guidelines.

Response 2: Thank you. We have aligned all the references accordingly.

Comment 3: Also, the survey questions are not presented in this study. Can the authors explain why this is not provided? A sample survey should be presented in the main text or in supplementary materials as a reference.

Response 3: The questions were provided in Methods. These are listed under Section 2.3 “Researcher-delivered statement and survey.”

Comment 4: Also, the limitation of the study is not provided in the discussion or conclusion section.

Response 4: The limitation section has already been delineated and occurs before the conclusion at Lines 369-382. “Limitations of the study include relatively small sample size drawn from health systems in the Northeast region of the US. While the characteristics of the two sites differed in terms of urban and rural populations, they both lie in liberal-predominant states. Inclusion of a more diverse geographic mix may have produced different results. Additional limitations include the lack of demographic data for participating patients. Keeping responses anonymous without identifying information allowed the elimination of informed consents which would have added significant time to the encounter and may have been more burdensome for hospitalized patients. The questions were kept very brief since respondents were hospitalized for acute conditions. However, a longer questionnaire may provide finer discrimination on patient perceptions. The survey questions developed by the researchers are not formally validated tools and future iterations of surveys may better reflect patient perceptions. Future studies should examine the long-term impact of these interventions, assessing whether follow-up education leads to sustained behavior changes, such as increased use of climate-related resources or other preventive measures, ultimately enhancing health outcomes in the face of climate-related risks.”

Comment 5: The authors claimed "A significant proportion of adult inpatients recognize the health risks associated with climate change, particularly in relation to air pollution and heat, often with an acute awareness of their personal vulnerability. However, while smartphone apps and other digital tools present potential opportunities for education and engagement, their effectiveness may be limited among certain patient populations, such as older adults or those with restricted access to technology". This claim seemed to be ordinary and common sense. Can the authors explained the novelty of their study?

Response 5: Thank you. This is the first inpatient study to assess climate health perceptions. We discussed in the paper that there was one prior study in the pediatric outpatient setting. This study differs in that it is with adult inpatients. The reviewer may be familiar with studies by the Yale Program on Climate Change Communication and George Mason Center for Climate Change Communication which has led large scale but generalized samples of US residents on climate change perceptions. There has been no study in this population. This was included in the Introduction, line 86-88: “While pediatricians have demonstrated success in integrating climate health education into well-child visits, as evidenced by positive results in parental perceptions, there remains a paucity of data concerning adult inpatient perceptions of climate-related health risks.[34]” We certainly want this to be clear to the reader and so have added this to the Discussion section: “As the first study of adult inpatient climate perceptions, the results of this study highlight that around half of hospitalized patients acknowledge awareness of environmental health impacts from air pollution and extreme heat.” Additionally, the use of applications to drive real-time adoption of preventive health measures in the setting of climate health concerns is an important finding because, while many patients understood environmental connections to their health, there were fewer who would interact with apps, suggesting that other modalities of communication should be considered. This was discussed Line 360-368.

Comment 6: This work seemed to be brief and not very comprehensive. With only 250 samples, can the authors explain if it is enough to conclude this study?

Response 6: As a pilot study of inpatient perceptions, there were no similar trials of this nature. However, we did note an outpatient study by Lewandowski et al where 234 respondents were assessed. This article was cited in the Methods section, lines 118-120. “The sample size was based on prior patient assessments of climate health perceptions, in which 234 families were interviewed.”

Comment 7: The authors should avoid using first-person tone (such as we) in scientific writing.

Response 7: Thank you. I have revised and removed all first-person references from the manuscript.

Comment 8: Also, PM2.5 should be written using a subscript.

Response 8: Thank you. I have corrected the subscripts.

Reviewer 2 Report

Comments and Suggestions for Authors

This is a significant and timely examination of the acceptability of mobile health (mHealth) resources for environmental surveillance among hospitalized adults—a population too often understudied in climate-health communication research. Quantitative and qualitative methods are well used by the authors to assess awareness and intention to use mobile applications to monitor air quality and heat. The study is well done, the manuscript is well written, and the findings are informative for environmental health professionals and clinicians.

But there are some areas that must be clarified, expanded on, or developed to increase the strength and impact of the manuscript. Specifically, the survey instrument has not been tested, demographic data are not reported, and more data on app use post-intervention would add to the study.

Abstract:

  • Consider adding the number of participants who lacked smartphone access in the abstract for context.

  • Clarify the proportion who accepted app installation assistance in the abstract.

Introduction

  • Consider briefly referencing the broader literature on digital health equity.

  • The statement comparing heat-related deaths to gun homicides is impactful but needs a citation in the introduction itself, not just later in the references.

Method:

  • Provide more details on how the standardized 30-second educational statement was developed and pilot-tested (if applicable).

  • Explain why demographic variables (e.g., age, gender, education, income) were not collected. Their absence limits the interpretability of differences in technology acceptance.

  • Clarify whether the app download assistance was attempted in real time or just offered.

Result:

  • Consider quantifying key themes in the qualitative results (e.g., how many participants mentioned technology barriers, sharing intent, etc.).

  • Add confidence intervals or chi-square values to site differences where applicable.

  • Address the relatively low number of qualitative comments (113 out of 250) as a limitation.

Discussion

  • Further elaborate on how these findings can be translated into practical inpatient interventions (e.g., environmental risk handouts, pre-discharge counseling).

  • Discuss potential bias introduced by interviewer-patient dynamics, especially if physicians conducted the surveys.

  • Include more on the potential for digital literacy training or offline alternatives for patients without smartphones.

Limitation

  • Add a note on self-selection bias (those more environmentally concerned may have been more likely to participate or respond favorably).

References

  • Double-check that all in-text references have full citations in the list. Some WHO links may need updating or archiving for permanence.

Additional Suggestions for the Authors

    • Consider adding a short follow-up survey in future research to evaluate actual app usage post-discharge.

    • Include a brief figure or schematic of the study design for visual clarity.

    • Explore the potential of extending the intervention to outpatient settings in future studies.

Comments on the Quality of English Language

It needs to improve.

Author Response

Response to Reviewer 2

This is a significant and timely examination of the acceptability of mobile health (mHealth) resources for environmental surveillance among hospitalized adults—a population too often understudied in climate-health communication research. Quantitative and qualitative methods are well used by the authors to assess awareness and intention to use mobile applications to monitor air quality and heat. The study is well done, the manuscript is well written, and the findings are informative for environmental health professionals and clinicians.

But there are some areas that must be clarified, expanded on, or developed to increase the strength and impact of the manuscript. Specifically, the survey instrument has not been tested, demographic data are not reported, and more data on app use post-intervention would add to the study.

Abstract:

  • Comment 1: Consider adding the number of participants who lacked smartphone access in the abstract for context.
  • Response 1: Thank you for the suggestion! We have incorporated this into the Abstract. We had a good deal of results and it helps to have these suggestions for what readers would be most interested in seeing highlighted in the Abstract.
  • Comment 2: Clarify the proportion who accepted app installation assistance in the abstract.
  • Response 2: Thank you for the suggestion! We have incorporated this into the Abstract as well.

Introduction

  • Comment 3: Consider briefly referencing the broader literature on digital health equity.
  • Response 3: Thank you for the suggestion. We have added the following section to the Discussion section: “In both states, ensuring digital health equity is essential, as individuals from lower socioeconomic backgrounds continue to face a persistent “digital divide” and significant barriers to adopting new technologies.[44, 45] Complacency in assuming widespread access neglects the reality that older adults, particularly those from economically challenged backgrounds, exhibit significantly lower engagement with digital health tools, often impacted by factors such as digital literacy and access to technology. [46, 47]
  • Comment 4: The statement comparing heat-related deaths to gun homicides is impactful but needs a citation in the introduction itself, not just later in the references.
  • Response 4: Apologies, we cited Drew Shindell’s work on heat-related deaths.

Method:

  • Comment 5: Provide more details on how the standardized 30-second educational statement was developed and pilot-tested (if applicable).
  • Response 5: We did not have these survey questions separately validated but have added the following section. To section 2.2 on methods: “The survey questions were developed by the authors and beta-tested with physicians outside the study with climate health expertise. The questions were kept intentionally short, in recognition of the acuity of patients’ condition in the hospital setting.” We mentioned this in the limitation section previously as well: “ The survey questions developed by the researchers are not formally validated tools and future iterations of surveys may better reflect patient perceptions.”
  • Comment 6: Explain why demographic variables (e.g., age, gender, education, income) were not collected. Their absence limits the interpretability of differences in technology acceptance.
  • Response 6: We agree that we cannot draw finer conclusions on patient characteristics. However, collecting this data would require a full IRB and an informed consent. We were concerned that the time spent going through a full informed consent would dissuade many hospitalized patients from continuing with the study and we aimed to keep this as minimally intrusive and long as possible.
  • Comment 7:
  • Response 7: Clarify whether the app download assistance was attempted in real time or just offered. Yes, my medical students, residents, and I helped our patients download the apps! We will spell this out in the Methods section. “If patients were interested in using the applications, the clinicians conducting the study helped the patients/families download them in real time.”

Result:

  • Comment 8: Consider quantifying key themes in the qualitative results (e.g., how many participants mentioned technology barriers, sharing intent, etc.).
  • Response 8: We appreciate the reviewer’s thoughtful suggestion. In this study, as the reviewer noted, fewer than half of participants (approximately 45%) chose to provide qualitative comments, and some individuals shared more than one. Because of this, quantifying responses as if they were representative of the entire cohort could be misleading. We therefore treated the qualitative data as exploratory, using it to enrich and contextualize the quantitative findings rather than to generate prevalence estimates. To acknowledge this explicitly, we have clarified in the Methods and Discussion that the qualitative themes highlight illustrative perspectives and patterns of response, but are not intended to be interpreted in a quantitative or generalizable manner. Relatively few patients (113 of 250) provided qualitative comments so qualitative assessment was drawn from a smaller pool of patient experiences. As a result, the qualitative themes highlight illustrative perspectives and patterns of response, but are not intended to be interpreted in a quantitative or generalizable manner. Future studies should examine the long-term impact of these interventions, assessing whether follow-up education leads to sustained behavior changes, such as increased use of climate-related resources or other preventive measures, ultimately enhancing health outcomes in the face of climate-related risks. Other studies should additionally explore deeper qualitative themes to effectively address potential gaps in care.”
  • Comment 9: Add confidence intervals or chi-square values to site differences where applicable.
  • Response 9: Thank you. We have added additional statistical results as follows: “A chi-square test comparing prior knowledge (“Were you previously aware...”) with patients' perception of their likelihood to use the app (“Do you think you would use...”) and their willingness to accept help installing the technology-based tool on their phone or tablet (“May I help you download...”) revealed a significant overall association (p = 0.001 and p = 0.004, respectively). However, when analyzing responses by location, no significant differences were observed at the Maine site (p = 0.457 and p = 0.685), suggesting that prior awareness did not influence responses in this group. In contrast, the New Jersey site showed a strong association between prior knowledge and both likelihood of use (p < 0.001) and willingness to accept help (p < 0.001), indicating that awareness played a more significant role in this population. These findings suggest that prior knowledge may not uniformly impact patient engagement but could be a relevant factor in specific settings.”
  • Comment 10: Address the relatively low number of qualitative comments (113 out of 250) as a limitation.
  • Response 10: Thank you. We included this as a limitation. (See above)

Discussion

  • Comment 11: Further elaborate on how these findings can be translated into practical inpatient interventions (e.g., environmental risk handouts, pre-discharge counseling).
  • Response 11: Thank you for this important point. We have expanded the Discussion to emphasize how these findings can inform practical inpatient interventions. Specifically, clew added this: “For individuals less inclined to engage with technology, clinicians could provide brief pre-discharge counseling on environmental risks, similar to other preventive health topics, and offer printed handouts with clear strategies for reducing exposure to poor air quality and extreme heat. Such materials could serve as accessible alternatives to smartphone applications. In addition, integrating environmental risk checks into routine discharge planning alongside medication and follow-up instructions may help ensure that patients leave the hospital with actionable, personalized guidance to protect their health. Strengthening public health warning systems for heat and poor air quality, and ensuring these alerts are disseminated in patient-centered ways, could further enhance the effectiveness of these inpatient interventions.”
  • Comment 12: Discuss potential bias introduced by interviewer-patient dynamics, especially if physicians conducted the surveys.
  • Response 12: Excellent point. The authors also recognize that interviewer–patient dynamics may have introduced bias, particularly as physicians or medical trainees conducted the surveys. Patients may have felt inclined to respond in a socially desirable manner or provide answers they perceived as preferred by the interviewer. To mitigate this, a standardized script was used for all encounters, responses were recorded immediately to reduce interpretation bias, and multiple coders independently reviewed qualitative comments to minimize individual influence.
  • Comment 13: Include more on the potential for digital literacy training or offline alternatives for patients without smartphones.
  • Response 13: We added the following section: For individuals less inclined to engage with technology, clinicians could provide brief pre-discharge counseling on environmental risks, similar to other preventive health topics, and offer printed handouts with clear strategies for reducing exposure to poor air quality and extreme heat. Such materials could serve as accessible alternatives to smartphone applications. In addition, integrating environmental risk checks into routine discharge planning alongside medication and follow-up instructions may help ensure that patients leave the hospital with actionable, personalized guidance to protect their health. Strengthening public health warning systems for heat and poor air quality, and ensuring these alerts are disseminated in patient-centered ways, could further enhance the effectiveness of these inpatient interventions.

Limitation

  • Comment 14: Add a note on self-selection bias (those more environmentally concerned may have been more likely to participate or respond favorably).
  • Response 14: Thank you. We added the following to our Limitation section. “The authors acknowledge the potential for self-selection bias in this study. Individuals who were already more environmentally concerned or interested in climate-health issues may have been more likely to participate in the survey or provide favorable responses. This could have resulted in an overestimation of awareness or willingness to engage with the proposed tools. However, participation was voluntary, brief, and unrelated to clinical care, and all eligible inpatients approached during the study period were included regardless of baseline views.”

References

  • Comment 15: Double-check that all in-text references have full citations in the list. Some WHO links may need updating or archiving for permanence.
  • Response 15: Thank you! I have updated and added additional citations as per another reviewer.

Additional Suggestions for the Authors

  • Comment 16: Consider adding a short follow-up survey in future research to evaluate actual app usage post-discharge.
  • Response 16: Thank you for the suggestion! We are considering how to further expand this initial pilot study.
  • Comment 17: Include a brief figure or schematic of the study design for visual clarity. Thank you. We created a visual abstract as requested.
  • Comment 18: Explore the potential of extending the intervention to outpatient settings in future studies.
  • Response 18: Other studies should additionally explore deeper qualitative themes to effectively address potential gaps in care in the inpatient and outpatient clinical settings.

Reviewer 3 Report

Comments and Suggestions for Authors

The manuscript titled "Hospitalized Adults’ Willingness to Use Mobile Apps for Air Quality and Heat Monitoring: A Survey-Based Study" looks interesting, but the study must demonstrate scientific soundness to be published in the journal.

1.The study should clearly identify the limitations of previous research on similar topics and explain how this work addresses those gaps to contribute new insights.

2.The study lacks a clear description of how the questionnaire was evaluated or validated to ensure it accurately captures the intended information.

3.The study does not explain the rationale for selecting the two study sites, which is important for interpreting and discussing the findings in context.

4.The discussion lacks comparison with similar existing studies, which limits the ability to contextualize the findings and may affect the credibility of the conclusions.

Author Response

Comment 1: The study should clearly identify the limitations of previous research on similar topics and explain how this work addresses those gaps to contribute new insights.

Response 1: Thank you. We have added and expanded our limitations section to address this concern. Limitations of the study include relatively small sample size drawn from health systems in the Northeast region of the US. While the characteristics of the two sites differed in terms of urban and rural populations, they both lie in liberal-predominant states. Inclusion of a more diverse geographic mix may have produced different results. Additional limitations include the lack of demographic data for participating patients. Keeping responses anonymous without identifying information allowed the elimination of informed consents which would have added significant time to the encounter and may have been more burdensome for hospitalized patients. The questions were kept very brief since respondents were hospitalized for acute conditions. However, a longer questionnaire may provide finer discrimination on patient perceptions. The survey questions developed by the researchers are not formally validated tools and future iterations of surveys may better reflect patient perceptions.

The authors acknowledge the potential for self-selection bias in this study. Individuals who were already more environmentally concerned or interested in climate-health issues may have been more likely to participate in the survey or provide favorable responses, which could have resulted in an overestimation of awareness or willingness to engage with the proposed tools. Participation was voluntary, brief, and unrelated to clinical care, and all eligible inpatients approached during the study period were included regardless of baseline views. The authors also recognize that interviewer–patient dynamics may have introduced bias, particularly as physicians or medical trainees conducted the surveys. Patients may have felt inclined to respond in a socially desirable manner or provide answers they perceived as preferred by the interviewer. To mitigate this, a standardized script was used for all encounters, responses were recorded immediately to reduce interpretation bias, and multiple coders independently reviewed qualitative comments to minimize individual influence. Relatively few patients (113 of 250) provided qualitative comments so qualitative assessment was drawn from a smaller pool of patient experiences. As a result, the qualitative themes highlight illustrative perspectives and patterns of response, but are not intended to be interpreted in a quantitative or generalizable manner. Future studies should examine the long-term impact of these interventions, assessing whether follow-up education leads to sustained behavior changes, such as increased use of climate-related resources or other preventive measures, ultimately enhancing health outcomes in the face of climate-related risks. Other studies should additionally explore deeper qualitative themes to effectively address potential gaps in care in the inpatient and outpatient clinical settings.

Comment 2: The study lacks a clear description of how the questionnaire was evaluated or validated to ensure it accurately captures the intended information.

Response 2: Thank you for the suggestion. We have added the following section to the Methods section. “The survey questions were developed by the authors and beta-tested with physicians outside the study with climate health expertise. The questions were kept intentionally short, in recognition of the acuity of patients’ condition in the hospital setting. If patients were interested in using the applications, the clinicians conducting the study helped the patients/families download them in real time.”

Comment 3: The study does not explain the rationale for selecting the two study sites, which is important for interpreting and discussing the findings in context.

Response 3: We have added the additional clarification for site selection. “The two sites were selected to increase geographic distribution and patient representation.” We acknowledge in the Limitation section that the study is limited to the Northeast US. These were chosen to facilitate rapid acquisition of data from a trusted colleague with availability to work on this project.

Comment 4: The discussion lacks comparison with similar existing studies, which limits the ability to contextualize the findings and may affect the credibility of the conclusions.

Response 4: Thank you for the comment. There are no other studies of inpatient perceptions of climate health impacts. The closest study, an outpatient study by Andrew Lewandowski, was frequently referenced throughout the paper. We have also placed this work in the context of larger population-based climate communication research. “This aligns with national trends, where a majority of Americans believe global warming is occurring, but suggests that those experiencing health issues may be more attuned to additional threats to their health. In Camden NJ, 77% of patients acknowledged the health impacts of environmental factors, which, while lower than the 90% in Maine, is still higher than in national studies. A nationwide survey found that 49% of individuals believed their families would be harmed by global warming, and 44% believed they themselves would be affected.[48] In the same sample, 70% of Americans agreed that global warming is happening, compared to only 13% who disagreed. This demonstrates a broad public recognition of climate change and suggests that ill patients may recognize environmental health impacts.”

Round 2

Reviewer 1 Report

Comments and Suggestions for Authors

The abstract is poorly written that separates all the section such as background, methods, results, and conclusions. it should uniformly integrated into a single paragraphs that include all components.

The use of first person (such as We) are still present in this work, which is not acceptable.

Although the authors give an explanation on the small sample size, it does not seemed to be convincing with only a 234 sample size. The authors should try to significantly increase the sample size to at least 500.

Section 2.3 presented the survey questions. It is strongly recommended to include the questions in a separate supplementary material section.

Can the authors please explain what is the novelty of this work? Because the methods and findings of this work seemed to be ordinary.

Author Response

Comment 1: The abstract is poorly written that separates all the section such as background, methods, results, and conclusions. it should uniformly integrated into a single paragraphs that include all components.

Response 1: Thank you. We have restructured the abstract.

Comment 2: The use of first person (such as We) are still present in this work, which is not acceptable.

Response 2: Apologies for missing this concern. I removed “we” throughout the document.

Comment 3: Although the authors give an explanation on the small sample size, it does not seemed to be convincing with only a 234 sample size. The authors should try to significantly increase the sample size to at least 500.

Response 3: We understand the reviewer’s concern. However as a pilot study, we wanted to prioritize getting this information out to clinicians in a timely manner and thus had to compromise on a large sample size. Additionally, this sample size is similar to the Lewandowski study which was conducted in the pediatric ambulatory setting.

Comment 4: Section 2.3 presented the survey questions. It is strongly recommended to include the questions in a separate supplementary material section.

Response 4: Because there were only 3 questions asked and many readers may not click into supplementary material, we thought it reasonable to include this in the body of the manuscript. One of the benefits of this intervention was the brevity and ease with which it can be integrated into an inpatient clinical encounter,

Comment 5: Can the authors please explain what is the novelty of this work? Because the methods and findings of this work seemed to be ordinary.

Response 5: The novelty lies in assessing an inpatient population which has not been previously studied for climate health impacts. The additional novelty is assessing if technology would be accepted by a vulnerable population of patients for climate risk avoidance.

Reviewer 3 Report

Comments and Suggestions for Authors

Thank you for submitting this manuscript. However, as a preliminary study, the content lacks sufficient scientific rigor and methodological soundness for publication in its current form.

I recommend the following improvements for future research:

  1. Unclear objectives and hypothesis: The study fails to clarify whether it aims to measure awareness or evaluate the effectiveness of the educational intervention.
  2. Change the study design to an intervention study with pre/post assessment and control group.
  3. No validation of the questionnaire: This might affect the accuracy and precision of the study. The questionnaire should be tested with a small group to validate its accuracy and precision before use.
  4. Develop a suitable and validated tool for evaluating climate health literacy.
  5. Increase the population or sample size and conduct stratified analysis.
  6. The authors should indicate the primary endpoint such as behavior change or knowledge improvement.

Author Response

Comment 1: Unclear objectives and hypothesis: The study fails to clarify whether it aims to measure awareness or evaluate the effectiveness of the educational intervention.

Response 1: The following objectives appear at the end of the Introduction. “This study assessed hospitalized adults’ perceptions of climate change as a health risk and their awareness of environmental factors affecting their well-being. Additionally, this study explores patients’ interest in using technology, such as smartphone applications, to monitor environmental risks and support personal health management.”

Comment 2: Change the study design to an intervention study with pre/post assessment and control group.

Response 2: This study is meant to serve as a foundation exploration, after which an intervention study may be conducted.

Comment 3: No validation of the questionnaire: This might affect the accuracy and precision of the study. The questionnaire should be tested with a small group to validate its accuracy and precision before use and develop a suitable and validated tool for evaluating climate health literacy.

Response 3: Thank you. We noted this in the paper. However, we had beta-testing with climate health experts prior to using the questions.

Comment 5: Increase the population or sample size and conduct stratified analysis.

Response 5: We agree a larger sample would be ideal. However as a pilot study, we wanted to prioritize getting this information out to clinicians in a timely manner and thus had to compromise on a large sample size. Additionally, this sample size is similar to the Lewandowski study which was conducted in the pediatric ambulatory setting.

Comment 6: The authors should indicate the primary endpoint such as behavior change or knowledge improvement.

Response 6: The study is not meant to assess behavior change but, rather, climate health perceptions among a hospitalized population. Additionally, we wanted to assess if phone applications would be a reasonable methodology for climate risk prevention.